# Preliminary Study of New Low-Temperature Hard Abrasion Resistant Fe-P and Fe-P-X (X = C or/and B) Casting Alloys

**DOI:** 10.3390/ma16103766

**Published:** 2023-05-16

**Authors:** Matija Zorc, Aleš Nagode, Jaka Burja, Borut Kosec, Milan Bizjak, Borut Zorc

**Affiliations:** 1Faculty of Natural Sciences and Engineering, University of Ljubljana, Aškerčeva 12, 1000 Ljubljana, Slovenia; 2Institute of Metals and Technology, Lepi pot 11, 1000 Ljubljana, Slovenia; 3Welding Institute Ltd., Ptujska 19, 1000 Ljubljana, Slovenia

**Keywords:** iron-phosphorus alloys, low melting temperature, as-cast state, high hardness

## Abstract

This article analyses the as-cast state of practically unknown Fe-P-based cast alloys with or without an addition of carbon and/or boron, cast into a grey cast iron mould. The melting intervals of the alloys were determined by DSC analysis, and the microstructure was characterized by optical and scanning electron microscopy with an EDXS detector. The hardness and microhardness of the alloys were also measured. Their hardness reached values between 52 and 65 HRC depending on chemical composition and microstructure, showing their high abrasion resistance. The high hardness is a consequence of the eutectic and primary intermetallic phases of Fe_3_P, Fe_3_C, Fe_2_B or mixed type. By increasing the concentration of metalloids and combining them, the hardness and brittleness of the alloys were increased. The alloys with predominantly eutectic microstructures were the least brittle. Depending on the chemical composition, the solidus and liquidus temperatures ranged from 954 °C to 1220 °C and were lower than those of the well-known wear-resistant white cast irons.

## 1. Introduction

High hardness is the basis for the high wear resistance of a material. For iron-based alloys, these properties reach the highest values in white cast irons. This is due to the high content of carbon and thus carbides in their microstructure [1]. Due to their brittleness, they are suitable for parts whose surface is exposed to small pieces of hard materials with low dynamic stresses. Such operating conditions are found during transport and the grinding and mixing of minerals and rocks in mines, quarries, cement plants, sand pits and concrete mixing plants. The wear-resistant parts are made either through the casting or surface welding of hard alloys onto the surface of non-alloy steel.

The chemical composition of white cast irons causes white fracture either over the entire thickness of the casting or in the surface area of thick-walled castings (from 10 to several 10 s of millimetres). The chemical compositions of various standardised white cast irons are in the following ranges (wt%): C = 2.0–3.7, Si ≤ 2.2, Mn ≤ 2, Cr ≤ 30, Mo ≤ 3.5, Ni ≤ 7, P ≤ 0.3 [1,2,3]. For example, the initial chemical compositions of malleable cast iron and unalloyed white cast iron are the same.

The microstructure of white cast irons depends on their chemical composition, crystallization rate and cooling rate. In hypoeutectic alloys, the microstructure is generally a mixture of eutectic carbides, Fe_3_C in unalloyed cast irons, (Cr,Fe)_7_C_3_ in alloyed cast irons and one or more products of transformed austenite (pearlite, bainite, martensite, retained austenite). In contrast, hypereutectic compositions in high-alloy cast irons cause primary carbides (Cr,Fe)_7_C_3_ to form as well. Their hardness depends on the microstructure of sand-cast non-alloyed irons; pearlitic white cast irons with a lower carbon content reach at least 320 HB, while those with a higher carbon content reach at least 400 HB [1]. Sand-cast alloyed white irons reach at least 450 HB (485 HV, 46 HRC) or 550 HB (600 HV, 53 HRC), depending on the chemical composition [2]. Casting in a mould increases the hardness of the alloy by at least 50 HB (60 HV, 3 HRC) [2]. After heat treatment, which removes high-temperature products of austenitic transformation from the microstructure, the hardness values of alloyed white cast irons typically reach 700–790 HB (800–950 HV, 62–67 HRC) [1] or at least 600 HB (660 HV, 56 HRC) for the first stage and 650 HB (715 HV, 59 HRC) for the second stage [2]. Wear-resistant cast irons with added vanadium and a low chromium content (e.g., C = 2.2–2.7, V = 6, Cr = 1; in wt%) achieve hardness in the range of 600–900 HV after heat treatment [4].

The filler metals for surfacing are also standardized [5], and the cast irons have the designations Fe14, Fe15 and Fe16. They can have a wide range of chemical compositions (wt%): C = 1.5–8, Mn ≤ 3, Cr = 10–40, Mo = 4–10, Ni ≤ 4, V ≤ 10, W ≤ 10, Nb ≤ 10, although other carbide-forming elements may also be added. Larger amounts of the carbide-forming elements Cr, Mo, V, W, Nb, Ti, Ta, Zr, Hf and B [4,5,6,7,8,9,10,11,12,13,14,15,16] are added to promote the formation of primary and special carbides of the above-mentioned elements that are present in the hard carbide eutectic matrix. The addition of Cr in surface-welded cast irons causes eutectic and primary carbides of the (Cr,Fe)_7_C_3_ type to predominate, while other carbide-forming elements promote the formation of tiny dispersed special carbides [17]. The more complex the chemical composition, the greater the number of different and complex hard phases in the microstructure. For example, borides and carboborides M_2_(C,B), M_3_(C,B), M_3_(C,B)_2_, M_7_(C,B)_3_ and M_23_(C,B)_6_ are found in high-alloy irons with high carbon and boron concentrations [13]. Their hardness exceeds 1500 HV [1,18,19], resulting in the microstructure that provides the best wear resistance. The hardness of the welded hard surface depends on the chemical composition as well as the number of welded layers and is in the range of 55–70 HRC [5,6]. Unalloyed cast irons contain only Fe_3_C, the hardness of which is lower than that of the previously mentioned special carbides of other carbide-forming elements. Different values for the hardness of Fe_3_C can be found in the literature: 700–800 HV [19,20], 740–960 HV for eutectic and 1070–1350 HV for primary Fe_3_C, with alloying further increasing its hardness [21].

In the last 25 years, hypoeutectic and hypereutectic Fe-B-based alloys containing 1–6 wt% B have been developed and tested [22,23]. The first group of alloys contained Cr additions in the range of 5–18 wt%. Later, cheaper alloys with little or no addition of alloying elements (C ≤ 0.5, Cr ≤ 3; wt%) were also produced. High hardness and wear resistance are attributed to Fe_2_B borides as well as Fe_3_(C,B) and Fe_23_(C,B)_6_ carboborides. The alloys can be modified with Ti, V and rare earth elements. This also leads to the formation of other hard phases in the microstructure, such as TiB_2_ and TiC. Both the modifiers and the subsequent heat treatment affect the size, shape and distribution of the borides, thus increasing the toughness of these alloys. The hardness values of Fe-B alloys range from 55 HRC to 62 HRC, depending on the state of the alloy and its chemical composition. Wear resistance increases with higher boron content and is comparable to white irons with a high Cr content.

Little information is available on Fe-P-type wear-resistant alloys, although there have been studies on electrochemical and chemical Fe-P surface deposits [24,25]. Surface layers formed by diffusion have a microhardness of 1000–1100 HV [26] and prove that Fe_3_P (P = 16 wt%) and Fe_2_P (P ≈ 22 wt%) phosphides are wear-resistant, hard, intermetallic compounds. They are comparable to the microhardness of heat-treated Ni-P surface layers with P = 1–13 wt% [27]. The vast majority of information on Fe-P alloys relates to grey cast iron. Fe_3_P is usually present in grey cast iron as a constituent of a usually undesirable brittle ternary eutectic called steadite (P = 6.89 wt%, C = 2.4 wt%; the microstructure at room temperature is Fe_3_P + Fe_3_C + ferrite) whose hardness reaches 700–800 HV [20]. It was found that a high concentration of P increases the steadite content in the microstructure and improves the wear resistance of grey cast iron [20,28]. Grey cast iron with continuous intergranular steadite formation exhibits the best wear resistance [20]. Since steadite increases the brittleness of the alloy, the phosphorus content is limited in all iron-based alloys. The highest P content is found in grey cast irons: in common grades, it is less than 0.6 wt% [20,29], while in those with good castability and wear resistance requirements, the P content is 0.6–1.8 wt% [20]. In other cast irons and steels, phosphorus is present in much lower amounts [1,30,31,32,33]. Although it is present in very small amounts, its characteristic segregations can cause problems [20,33,34] and make steel sheets brittle [34]. The main reason that Fe-P-based casting alloys with higher phosphorus content have not been explored further, despite their promise as cheap wear-resistant materials, is their brittleness. For this reason, wear-resistant composites of Fe-P-B alloys with a ductile matrix based on copper alloys have been developed [35]. However, it is worth noting that Fe-P-X alloys (X = metalloids) are also used as metallic glasses with excellent soft magnetic properties [36].

This article discusses the as-cast state of Fe-P and Fe-P-X alloys (X: C, B; individually or in combination) cast in a permanent mould. The alloying elements P, C and B form hard intermetallic compounds with Fe that provide a basis for good wear resistance. The binary Fe-C and Fe-B alloys are also analysed for comparison.

## 2. Scientific Background

Alloys with high wear resistance always have hard intermetallic compounds in their microstructure. These include carbides, borides and phosphides. The higher the content of hard intermetallic compounds, the better the wear resistance. In (Fe,Cr)-C-based alloys, good wear resistance is achieved by eutectic and primary special carbides (Cr,Fe)_7_C_3_. If other carbide-forming elements are added, other special carbides also form in the microstructure of (Fe,Cr,X)-C alloys (X = Mo, V, W, Nb, Ti, Ta, Zr, Hf). These alloys are characterized by high production costs due to the expensive raw materials.

Fe-P-based alloys with or without the addition of C and/or B have two positive properties compared to known wear-resistant (Fe,Cr)-C-based alloys. The first is the approximately twofold lower price of the raw materials required for their production, or even lower if compared with alloys containing not only Cr but also other carbide-forming elements in their chemical composition. The second characteristic of Fe-P alloys is their low eutectic temperature (*T*_E_ = 1048 °C for Fe-Fe_3_P alloys with P = 2.8–16 wt%), while the melting range of alloys with P = 8–15 wt% is *T* = 1048–1166 °C [37]. The eutectic temperature of Fe-Fe_3_P alloys is 100 °C lower than that of Fe-Fe_3_C alloys and 125 °C lower than that of Fe-Fe_2_B alloys [37]. The addition of C and B in appropriate amounts further lowers the eutectic temperature and narrows the melting range of Fe-Fe_3_P alloys, as these two elements lower the melting temperature of Fe. An example of this is a ternary eutectic steadite with eutectic temperature *T*_E_ = 950 °C [20]. All known Fe-based wear-resistant alloys have higher eutectic temperatures: those alloyed with Cr and Ni (*T*_E_ ≈ 1200 °C) and those alloyed with Cr (*T*_E_ ≈ 1230–1270 °C) [38]. This also means that these alloys melt in a temperature range of *T*_S-L_ ≈ 1200–1350 °C, depending on their composition [38]. The lower melting temperatures of Fe-P and Fe-P-X (X = C or/and B)-based alloys also mean that less energy is consumed per unit of alloy produced. Consequently, this means both lower production costs and a lower carbon footprint. Since P significantly improves castability, these alloys should also be easy to cast into more complex shapes.

## 3. Experiments

### 3.1. Production of Alloys

The alloys were produced in a powerline-frequency induction furnace with an aluminate coating of the crucible (an exception was Fe-C white cast iron intended for tempering, which was extracted from the casting channel). The mass of the batch was 5 kg of each different alloy and consisted of non-alloy steel S235, ferroalloys FeP (17.65 wt% P), FeB (17.0 wt% B) and carbon for nodular cast iron, individually or in combination, depending on the alloy. Ar 4.6 quality was blown onto the melt. The casting temperature was 1350 °C for the Fe-C alloy, 1100 °C for the Fe-P-C alloy and 1250 °C for the others. It was measured by IR Thermometer Optris CTlaser 05M: 1000–2000 °C/525 nm/150:1/1 ms/FF 24 mm@. The melt was poured into a vertical permanent mould made of grey cast iron, the walls of which were 22, 25 and 32 mm thick, coated with zirconia and preheated to T = 100 °C. The casting had a rectangular cross-section of 40 mm × 20 mm and was 300 mm long.

### 3.2. Research Methods

The chemical compositions of the alloys were determined by ICP OES and carbon combustion (Eltra combustion technique apparatus) methods. On the basis of the properties determined in the as-cast state (melting ranges, microstructure, hardness), the usefulness of these alloys and the meaningfulness of further in-depth investigations could be assessed. The melting ranges were determined by DSC analysis with a heating and cooling rate of 10 °C/min (Jupiter-NETZSCH STA 449 C device, Waltham, MA, USA). The mass of the samples was between 0.20 g and 0.30 g. One DSC analysis was performed for each alloy. The sample was taken from the middle of the central part of the casting since average solidification conditions are expected at this location. The microstructure was studied using a Leitz Laborlux 12 ME optical microscope (Stuttgart, Germany) and a FEG SEM Thermo Fisher Quattro S electron microscope (Waltham, MA, USA) with an Oxford Instruments Ultim^®^ Max EDXS SDD detector (Abingdon, UK), which was used to analyse the distribution of chemical elements in the microstructural phases (mapping) and their chemical compositions. Carbon cannot be accurately measured quantitatively, but differences in its distribution can be determined qualitatively by mapping. Thus, assumptions can be made about the presence of cementite and (P,C), (B,C) intermetallic phases. The metallographic samples were wet-ground with SiC abrasive paper with a grit size of up to #2400 and polished with diamond paste with a grit size of 1 μm. All samples were etched with 4% Nital, except for the Fe-P-B(2), which was experimentally etched with Vilella’s reagent (ethanol with 1% picric acid and 5% hydrochloric acid). The macro hardness HV10 of the alloys was measured using a Swiss Max 300 device (Mississauga, ON, Canada). The measurement uncertainty of the instrument was within the measurement uncertainty range of the control plate ± 7 HV. Hardness was measured according to the standard EN ISO 6507-1:2018. For each alloy, 10 measurements were made, from which mean values and standard deviations were calculated. The microhardness HV0.3 and HV0.5 of various microstructural constituents was measured using the Shimatzu microhardness tester type M (Kyoto, Japan). The HRC values are given next to the measured HV10 values. Because of the intermetallic phases, they are comparatively determined from the table for Cr-alloyed white cast iron [1]. The values are approximate because the HV→HRC converters were different for different materials [1,19].

## 4. Results and Discussion

### 4.1. Chemical Compositions and Melting Ranges

The chemical compositions of the studied alloys are listed in Table 1. All experimental alloys had a high phosphorus content and different boron and carbon contents. Due to rapid cooling in the permanent mould, the alloys were not in an equilibrium state. Therefore, the DSC heating curves were applicable for the analysis of the as-cast state. The melting ranges and their shapes differed from each other due to the different chemical compositions and microstructures of the alloys. All experimental alloys with high phosphorus content had lower solidus and liquidus temperatures than various commercial white cast irons (Table 1; [38] Section 2 of this article).

The number and intensity of endothermic reactions (or exothermic on cooling) in DSC curves were proportional to the number of different microstructural constituents. Some alloys had only one strongly pronounced eutectic reaction, but there was a possibility that a second reaction was present that was either not visible or barely visible on the heating curve due to a small quantity of the phase that was either dissolving in the eutectic melt or melting at a slightly higher temperature than the main phase (Figure 1a). The device nevertheless detected the melting of such a phase. For other alloys, the reactions were more strongly pronounced (Figure 1b). The lowest temperature was always a low-temperature eutectic reaction, while reactions occurring at higher temperatures were either higher-temperature eutectic reactions or reactions of various primary phases.

### 4.2. Microstructure and Hardness of As-Cast State at High Cooling Rates

Due to rapid cooling in the permanent mould, the alloys were not in an equilibrium state. The microstructure of the alloys varied considerably due to the different chemical compositions. Fe-C and Fe-P-B(3) alloys were hypoeutectic and their microstructure consisted of a eutectic and a primary solid solution. Other alloys were hypereutectic, indicated by sharp-edged and acicular primary intermetallic phases in the microstructure. The eutectics in all alloys consisted of transformed or untransformed Fe-based solid solution and Fe-based intermetallic phases of Fe_3_P (16 wt%), Fe_3_C (6.7 wt%), Fe_2_B (8.8 wt%) [37] or mixed type, depending on the chemical composition. The figures show the impressions of the HV10 hardness measurements.

The well-known microstructure of the hypoeutectic Fe-C alloy consisted of binary eutectic (pearlite + Fe_3_C) and fine-lamellar pearlite (Figure 2). Pearlite consists of 0.76 wt% C, while ledeburite consists of 4.3 wt% C. The difference is clearly visible in the distribution of carbon. Although the EDXS method is not intended for the determination of carbon, the mapping of this alloy demonstrates the applicability of the said method for the qualitative determination of the carbon distribution in the microstructure.

The microstructure of the Fe-B alloy consisted of a binary eutectic (*α* + Fe_2_B; measured values of B = 4.04–4.77 wt%), which also contained some primary intermetallic Fe_2_B phase (measured values of B = 9.63 wt%), (Figure 3).

The Fe-P(1) alloy exhibited a microstructure consisting of a binary eutectic (*α* + Fe_3_P; measured values of P = 10.52 wt%), in which some primary intermetallic Fe_3_P phase was also found (measured values of P = 15.5 wt%) (Figure 4a). The latter was the main microstructural constituent of the Fe-P(2) alloy. Some intergranular eutectic was also visible in this alloy (black lines, Figure 4b).

The microstructure of the Fe-P-C alloy consisted of fine ternary eutectic steadite (Fe_3_P + Fe_3_C + ferrite; measured value of P = 6.82 wt%) and larger white phases in the form of irregular shapes, needles and very small square shapes (Figure 5). The last one mentioned was the primary intermetallic phase Fe_3_P.

The uneven white shapes had a phosphorus content similar to that of the clearly visible ternary eutectic (measured value of P = 6.99 wt%). As indicated by the EDXS mapping analyses, phosphorus was present as a part of the eutectic in the form of Fe_3_P coarse needles or lamellae. Among them were lamellae and needles that were rich in carbon. Some thin, long needles looked like a primary Fe_3_C. A barely detectable transformation at T = 972 °C on a DSC heating curve was likely a binary eutectic (Fe_3_P + Fe_3_C) that was not etched by Nital, but may have also been an anomalous ternary eutectic steadite.

The microstructure of the Fe-P-B(1) alloy consisted of a (α + Fe_3_(P,B)) eutectic [35] (measured values of P = 8.1–9.0 wt% and B = 0.67–1.1 wt%) and a white intermetallic Fe(P,B) phase (measured values of P = 6.63–7.56 wt% and B = 3.0–3.54 wt%). Fine particles of Fe_2_B intermetallic phase were also present in the latter, which the EDXS mapping analysis confirmed in the Fe-P-B(2) alloy as well. The microstructure is shown in Figure 6.

The microstructure of the Fe-P-B(2) alloy consisted of two intermetallic phases (Figure 7). One was a phosphorus-rich lighter matrix (measured values of P = 13.1 wt%) with some boron in it (measured values of B = 1.15 wt%), indicating that this phase was Fe_3_(P,B). The other was a boron-rich darker phase (measured values of B = 9.6 wt%) that also contained a tiny amount of phosphorus (measured values of P = 0.16 wt%). This phase was Fe_2_B.

The microstructure of the Fe-P-B(3) alloy (Figure 8) consisted of a (α + Fe_3_(P,B)) eutectic (measured values of P = 9.42–11.09 wt% and B = 0.6–0.72 wt%) and a small amount of Fe-based primary non-equilibrium solid solution (measured values of P = 2.63–4.0 wt%). Due to such a microstructure, the DSC heating curve (Figure 1a) showed only one strongly pronounced endothermic peak. This also included the melting reaction of the small amount of solid solution at T ≈ 1042 °C. The dendrite shown in the EDXS analysis consisted of three distinct colour segments. The light part was a solid solution, while the darker parts were iron and phosphorus oxides.

Due to their similar chemical compositions, the microstructures of Fe-P-B-C(1), Fe-P-B-C(2) and Fe-P-B-C(3) alloys were very similar (Figure 9a–c). Etching with Nital showed only the predominant white phase with a small amount of eutectic (consisting of ferrite or pearlite and intermetallic phase). In reality, the microstructure of these alloys was complex, as shown by the mapping analysis of the Fe-P-B-C(2) alloy (Figure 9d). The white matrix consisted of P and B (measured values of P = 10.62 wt% P and B = 1.1 wt%) and was Fe_3_(P,B). Grey needles of Fe_2_B (measured values of B = 9.24 wt%) were visible in the matrix, as well as light grey, randomly shaped, carbon-rich regions that also contained some boron (Figure 9d). These appeared to be eutectic cells (Fe_3_C + Fe_2_B).

The microstructure of the Fe-P-B-C(4) alloy consisted of a much higher proportion of eutectic (measured values of P = 11 wt%, B = 0.45 wt%, +C) than the other three alloys of the same type (Figure 10). The white phase (measured values of P = 12.77 wt% and B = 0.40 wt%) was Fe_3_(P,B). At the edges of the white phase and within the eutectic there was also a small quantity of grey, elongated, carbon-rich phase with a boron content of 1.86 wt%. Due to such a microstructure, the DSC curve (Figure 1b) had a strongly expressed low-temperature endothermic eutectic melting peak and a smaller high-temperature melting peak of the intermetallic phase.

Due to primary and eutectic intermetallic phases (iron phosphides, borides, carbides and mixed type), all four alloys were very hard (Table 2). Fe-P-B-C(4) had the lowest hardness, which was due to the high content of eutectic in the microstructure.

The Fe-C, Fe-B, Fe-P(1) and Fe-P-B(3) alloys with predominantly eutectic microstructures were the least hard. Their average hardness did not exceed 600 HV (53 HRC). Of these, the hypoeutectic Fe-C alloy achieved the lowest hardness, which was due to a higher content of fine-lamellar pearlite. The ternary eutectic Fe-P-C alloy had a much higher hardness than other predominantly eutectic alloys.

Hardness increased with a higher content of primary intermetallic phases, which was clearly evident when comparing alloys of the same group. In general, the hardness of these alloys exceeded 700 HV (58 HRC). The hardest alloy was Fe-P-B-C(1), with an average hardness of 875 HV (65 HRC), while individual areas exceeded 900 HV (66 HRC). The variations in the measured hardness in the 38–203 HV range were a consequence of the different microstructural constituents and the as-cast state of the alloys. They were smaller in binary and predominantly eutectic alloys (38–46 HV) than in alloys with many constituents (111–203 HV). This was a logical consequence of a more uniform distribution of the fine constituents of the eutectic microstructure. The hardness of binary Fe-P alloys increased with higher P content, which was due to a larger amount of the primary intermetallic phase Fe_3_P in the microstructure. Despite a lower P content compared to the eutectic Fe-P(1) alloy (by approximately 3 wt%), the hardness of Fe-P alloys could be maintained with the addition of approximately 0.7 wt% B. With the addition of approximately 2.3 wt% C or B, the hardness increased on average by approximately 135 HV. With the simultaneous addition of B and C to the Fe-P alloy (approximately 0.6–0.75 wt% each), the hardness increased by approximately 110 HV. An even larger increase of approximately 230 HV could be achieved by adding 1.2–1.6 wt% C and 1.3–2.6 wt% B. This increase was due to the formation of various intermetallic phases that were often of mixed type. The microhardness values prove that iron borides are harder than iron phosphides and iron carbides.

Due to the intermetallic phases, the alloys were also very brittle. This was evidenced by radial cracks propagating from the corners and sides of the Vickers hardness indentations. This is also characteristic of ceramic materials and glasses [39,40]. The longer the cracks and the more cracks there are, the more brittle the material is and the lower its fracture toughness [40]. Primary intermetallic phases were much more brittle than eutectic microstructures, which consisted of a solid solution and an intermetallic phase. A ductile solid solution in the eutectic reduced the brittleness of the microstructure. Therefore, no cracks were found at the corners of the hardness indentations in Fe-P(1) and Fe-P-B(3) alloys.

### 4.3. DSC Analysis and Microstructure

The consequence of rapid cooling in the permanent mould was a non-equilibrium microstructure. This was shown by comparing the DSC heating curves (crystallisation at a high cooling rate in the mould) and the cooling curves (remelting and crystallisation of the DSC sample at a low cooling rate) of Fe-P-C and Fe-P-B(1) alloys and their microstructures (Figure 11 and Figure 12).

The heating and cooling DSC curves of Fe-P-C alloys were almost identical. The difference was observed at slightly lower solidification temperatures during the cooling of the DSC sample (Figure 11a,b). The microstructure of the Fe-P-C alloy in the rapidly cooled as-cast state (the microstructure was described in the previous section) was non-uniform with a much finer ternary eutectic, in contrast to the uniform microstructure of the remelted and slowly cooled homogenized alloy. The alloy was slightly hypereutectic on the Fe_3_P side, as confirmed by individual, very small square Fe_3_P phases in the ternary eutectic of the slowly cooled DSC analysis sample.

The DSC heating and cooling curves of the Fe-P-B(1) alloy were different and consistent with the microstructure (Figure 12a,b). The different DSC curves clearly show the non-equilibrium microstructure of the rapidly crystallised state in the mould and the equilibrium microstructure of the slowly crystallised state in the DSC sample. In general, the microstructure of both states consisted of the (α + Fe_3_(P,B)) eutectic and white intermetallic phases. The latter ones were not the same in the as-cast state and slowly cooled DSC sample. In the as-cast state, the white intermetallic phases were Fe-(P,B) with a measured chemical composition of P = 6.63–7.56 wt% and B = 3.0–3.54 wt%. In the slowly cooled DSC sample, the intermetallic phases were Fe_2_B with a measured chemical composition of B = 10.3–10.53 wt%. In the as-cast state, the content of hard primary phases was much higher, which is clearly reflected in the DSC curve, while the eutectic was much finer than in the slowly cooled DSC analysis sample. The solidification temperatures were lower on the cooling curve in this case as well. It is clear that homogenisation and cooling rate have a decisive influence on the microstructure and solidus and liquidus temperatures of alloys and thus presumably on the mechanical properties of the alloys studied. Therefore, it would make sense to investigate this topic further in the future.

At this stage of the research, 4% Nital was used as an etchant. An experiment with Vilella’s reagent showed that other known etchants could be experimented with in the future as well. A 4% Nital etched only the ferrite phase, clearly revealing ferrite and pearlite in the eutectic, while intermetallic phases remained unetched and white. Nital thus clearly revealed all the microstructural constituents of hypoeutectic and hypereutectic alloys with one primary phase. However, it showed no difference between various intermetallic phases. White-looking phases may consist of two or more different intermetallic phases of primary or even eutectic type (example: Fe_3_P and Fe_3_C in steadite were not specifically expressed after etching with Nital; potentially present eutectic cells of the binary eutectic Fe_3_P + Fe_3_C, therefore, remained hidden). The shortcomings of Nital can be complemented with a mapping analysis of the white unetched areas. However, these shortcomings also have a bright side, as the proportion of the white areas can serve as a basis for determining (1) an overall proportion of hard wear-resistant intermetallic phases in the microstructure and (2) a proportion of a visible eutectic (consisting of ferrite or pearlite and an intermetallic phase) or a proportion of coarse intermetallic phases in the form of larger white areas that do not show a eutectic microstructure after etching with Nital (these are primary phases or eutectics consisting solely of intermetallic phases). It is clear that the greater the proportion of large white areas in the microstructure, the higher the hardness and wear resistance.

## 5. Conclusions

The results of the preliminary study show that the melting ranges of Fe-P-based alloys are at lower temperatures compared to commercial types of white cast iron. The results also show that Fe-P-based alloys cast in a permanent mould and cooled rapidly have a very high hardness. Depending on the amount and type of alloying elements and the type of microstructure, hardness values reach 52–65 HRC. The basis for good abrasion resistance is thus assured.

The microstructure of all alloys consists of a eutectic and primary phosphides. If C and B are added, iron carbide and iron borides or intermetallic phases of mixed type are also present in the microstructure. The higher the content of intermetallic phases (primary phases or eutectics consisting solely of intermetallic phases), the higher the hardness and brittleness of the alloys. Carbon and boron increase the hardness of Fe-P alloys. Iron boride is harder than iron phosphide and iron carbide.

The high brittleness of alloys cast in a permanent mould and cooled rapidly is considerable; therefore, they are probably less suitable for practical use in these states. The difference in brittleness of the individual phases in the alloys has a strong influence on their wear resistance. If microcracks initiate in very hard intermetallic phases, there is a high probability of particles tearing from the surface due to the action of the abrasive. This is especially true when the particles impact the surface or when the moving particles are between two surfaces of different hardness. For practical applications, it is, therefore, necessary to obtain the least brittle state of the alloys studied. That the microstructure is strongly dependent on the cooling rate is clear from the comparison of cast alloys and slowly cooled alloys of DSC samples. The microstructure of cast alloys is not in an equilibrium state, while slowly cooled DSC samples have an equilibrium microstructure. Thereby, the proportions and size of the individual microstructure constituents change. It is clear that this also changes the mechanical properties and brittleness of the alloys. For the practical use of casting alloys, it will be necessary to find the least brittle state. In the as-cast state in which they were analysed, they are at best useful in the manner described in the reference [35]. For this reason, the effect of different solidification and cooling rates in the solid state on the microstructure, hardness, brittleness and wear resistance of these alloys, as well as the effect of homogenisation on the above-mentioned properties, should be studied in the future. Accurate and very detailed characterisation of the microstructural constituents and their distribution will only make sense for the least brittle, practically usable state of the alloys.

## Figures and Tables

**Figure 1 materials-16-03766-f001:**
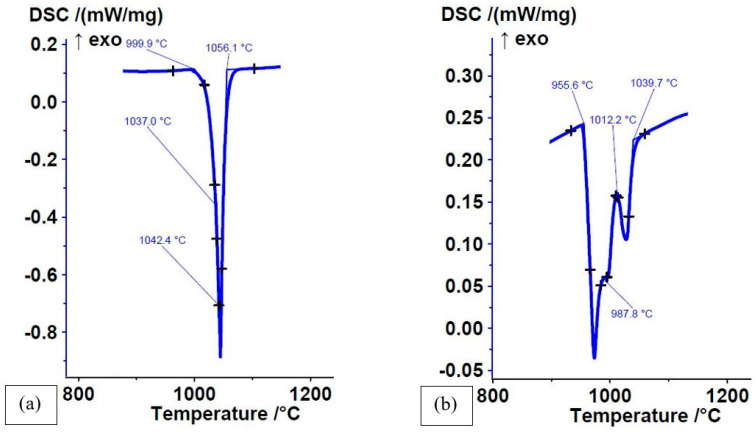
DSC heating curves of as-cast state at high cooling rates: (**a**) Fe-P-B(3) alloy; ΔH = −80.89 J/g); (**b**) Fe-P-B-C(4) alloy; ΔH = −71.44 J/g.

**Figure 2 materials-16-03766-f002:**
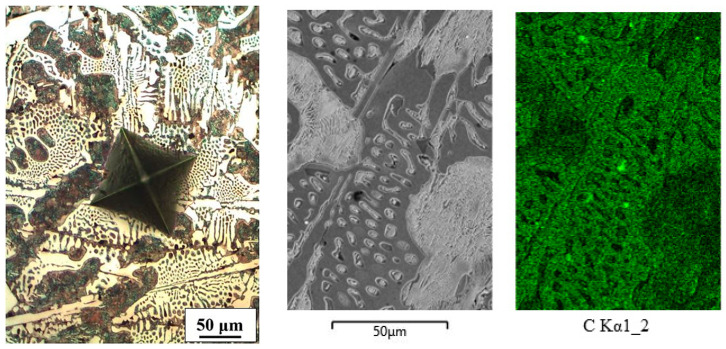
Microstructure and mapping of Fe-C alloy (white spots on C mapping are pores).

**Figure 3 materials-16-03766-f003:**
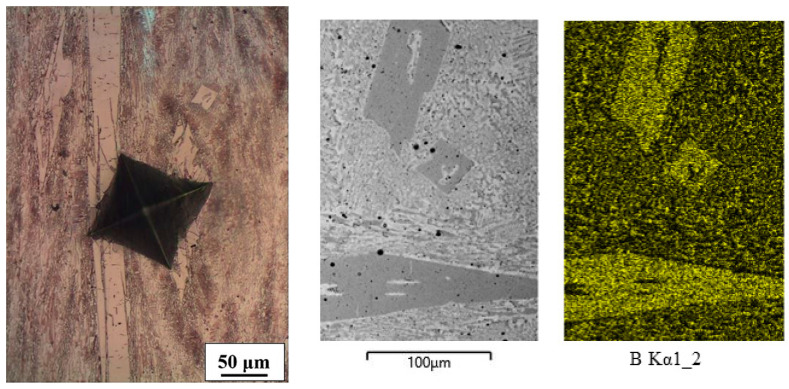
Microstructure and mapping of Fe-B alloy.

**Figure 4 materials-16-03766-f004:**
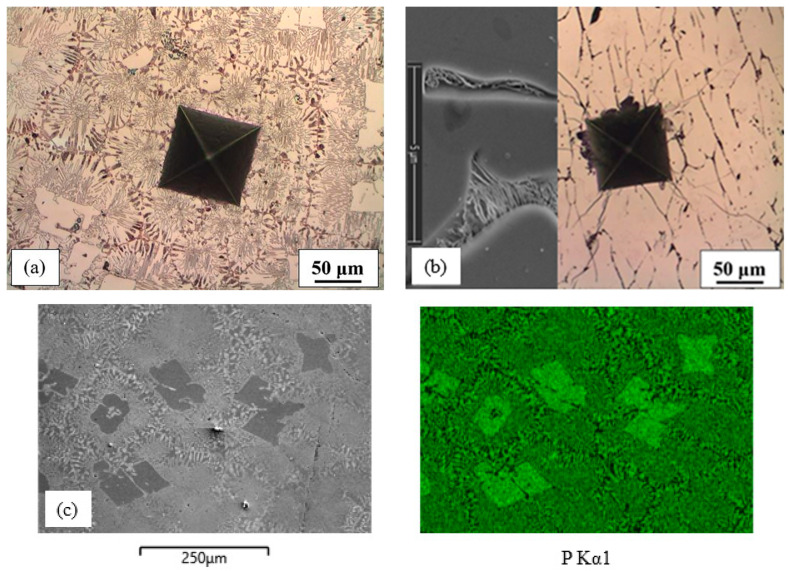
Microstructure and mapping of Fe-P alloys: (**a**) Fe-P(1); (**b**) Fe-P(2), eutectic at Fe_3_P grain boundaries; (**c**) mapping of Fe-P(1) alloy.

**Figure 5 materials-16-03766-f005:**
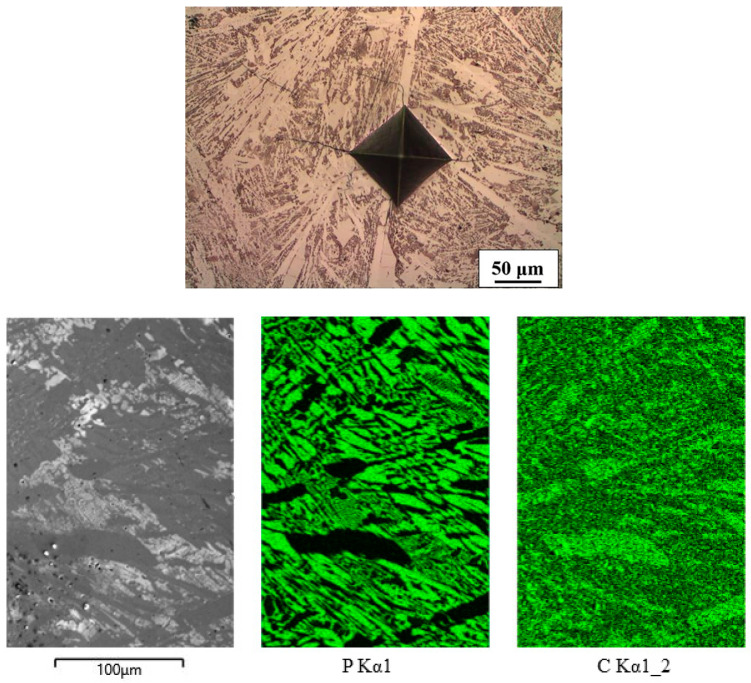
Microstructure and mapping of Fe-P-C alloy.

**Figure 6 materials-16-03766-f006:**
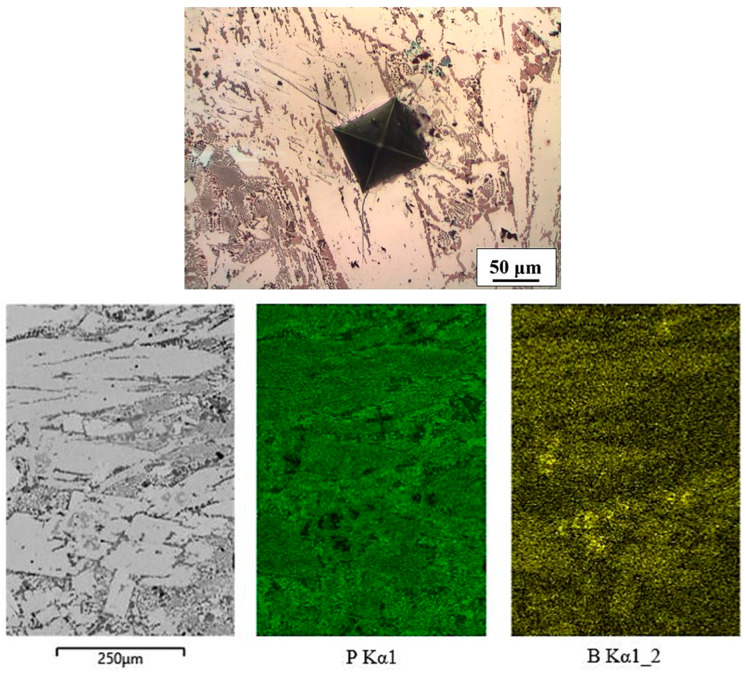
Microstructure and mapping of Fe-P-B(1) alloy.

**Figure 7 materials-16-03766-f007:**
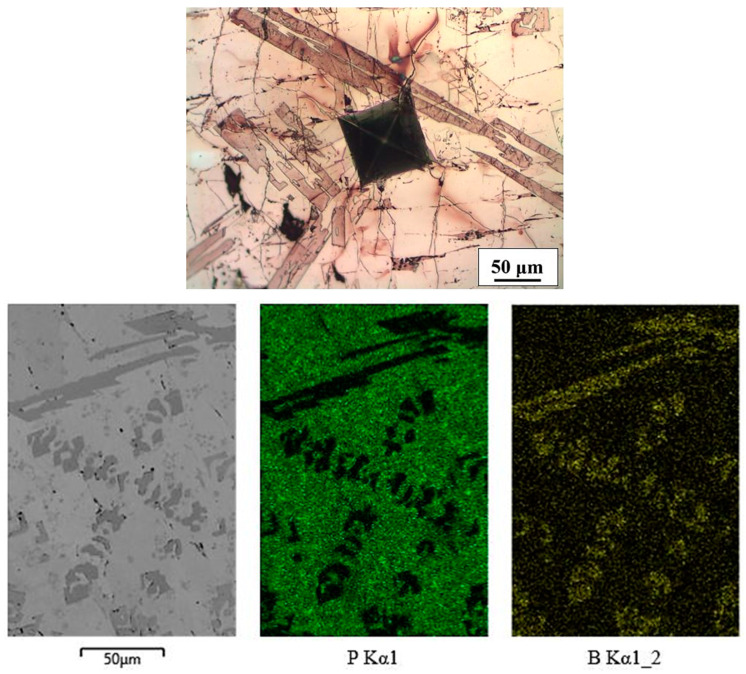
Microstructure (Vilella’s reagent) and mapping of Fe-P-B(2) alloy.

**Figure 8 materials-16-03766-f008:**
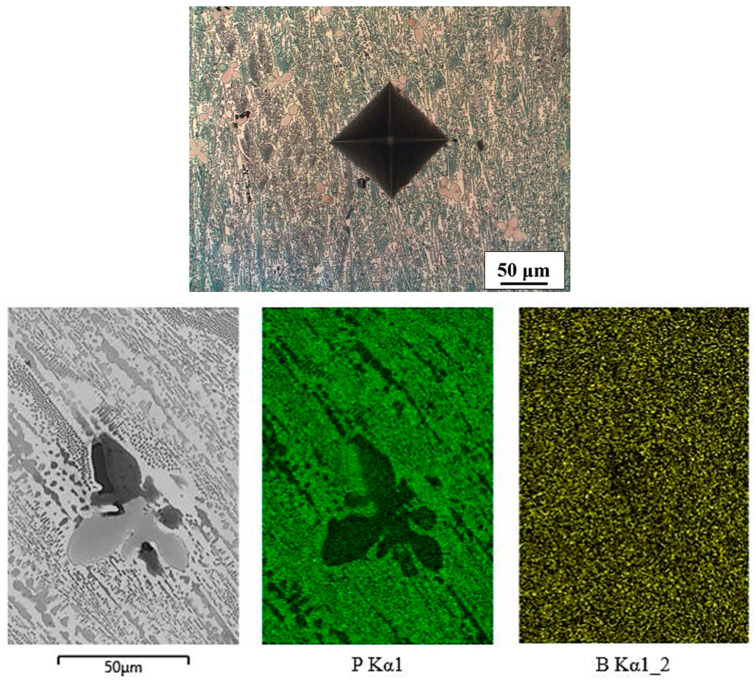
Microstructure and mapping of Fe-P-B(3) alloy.

**Figure 9 materials-16-03766-f009:**
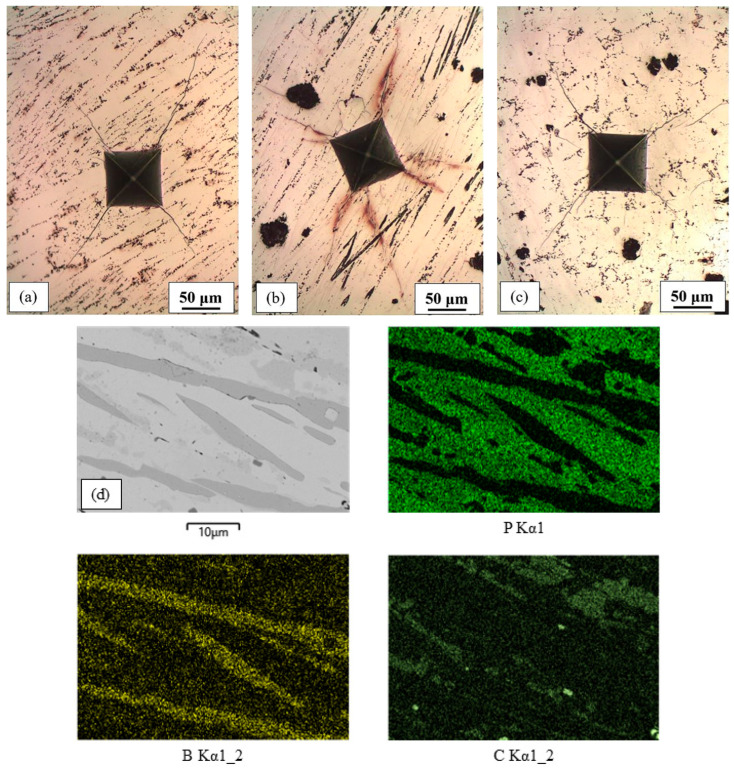
Microstructure of Fe-P-B-C alloys: (**a**) Fe-P-B-C(1); (**b**) Fe-P-B-C(2); (**c**) Fe-P-B-C(3); (**d**) mapping of Fe-P-B-C(2).

**Figure 10 materials-16-03766-f010:**
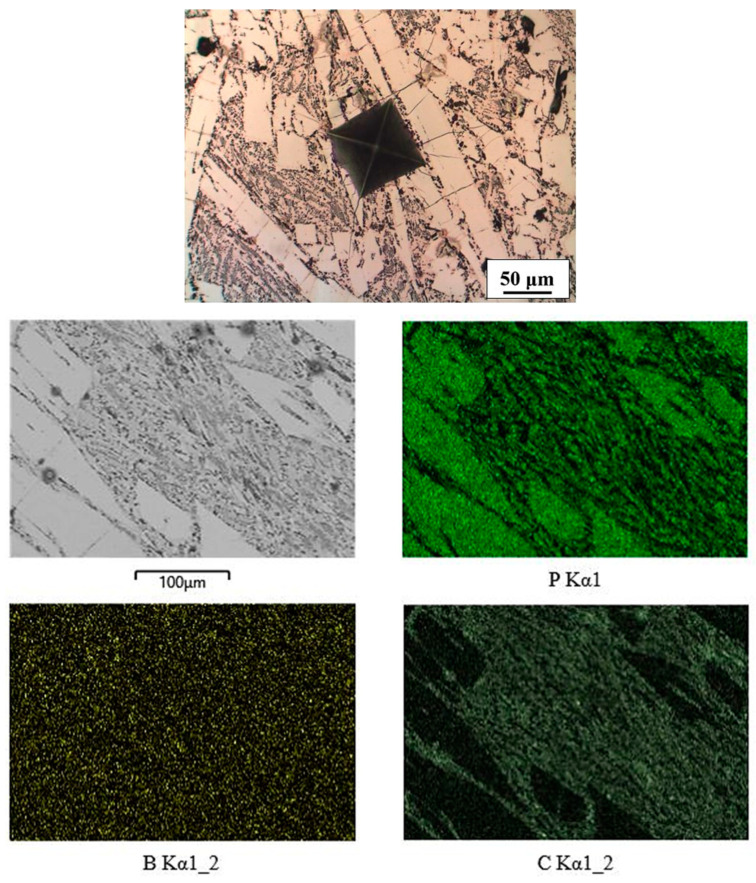
Microstructure and mapping of Fe-P-B-C(4) alloy.

**Figure 11 materials-16-03766-f011:**
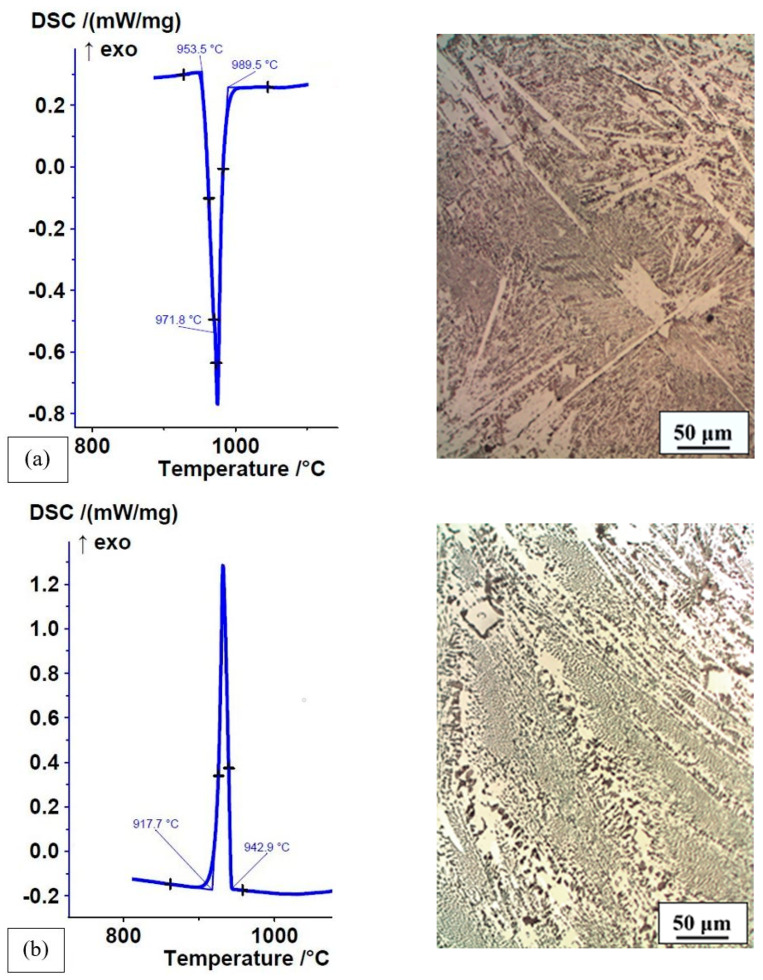
DSC curves and microstructure of Fe-P-C alloy: (**a**) heating curve of as-cast state with crystallisation at a high cooling rate in the permanent mould, ΔH = −105.6 J/g; (**b**) cooling curve of as-cast state with remelting and crystallisation at a low cooling rate of DSC sample, ΔH = 109.4 J/g.

**Figure 12 materials-16-03766-f012:**
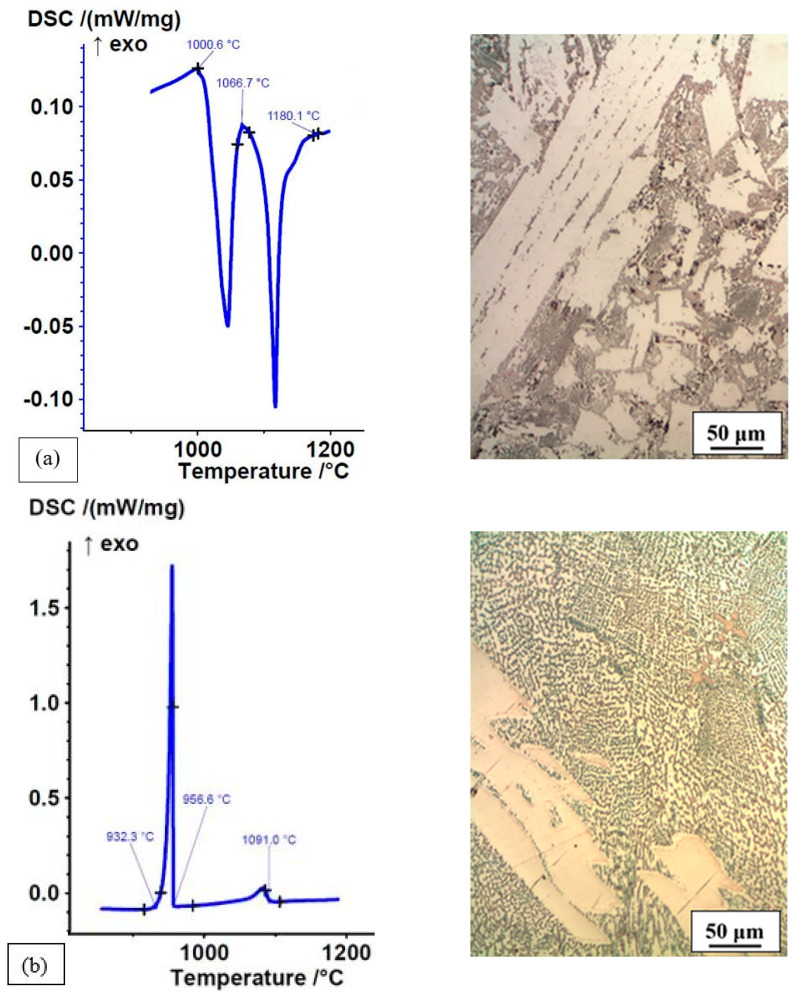
DSC curves and microstructure of Fe-P-B(1) alloy: (**a**) heating curve of as-cast state with crystallisation at a high cooling rate in the permanent mould, ΔH = −60.8 J/g; (**b**) cooling curve of as-cast state with remelting and crystallisation at a low cooling rate of DSC sample, ΔH = 105.4 J/g.

**Table 1 materials-16-03766-t001:** Chemical compositions (wt%) and melting ranges TS-L of as-cast state.

Alloy ^#^	Fe	Mn	P	B	C	Si	≈T_S-L_ (°C)
Fe-C	rest	0.66	0.015	/	2.95	0.82	1137–1290
Fe-B	rest	1.24	0.51	4.3	0.11	0.26	1110–1170
Fe-P(1)	rest	0.76	10.4	/	0.11	0.19	1050–1070 *
Fe-P(2)	rest	1.23	14.9	/	0.05	0.08	1035–1160 *
Fe-P-C	rest	1.57	7.4	/	2.20	0.34	954–990
Fe-P-B(1)	rest	0.77	7.0	2.1	0.08	0.21	1012–1180
Fe-P-B(2)	rest	1.40	9.3	2.5	0.09	0.21	1053–1220
Fe-P-B(3)	rest	0.57	7.6	0.73	0.07	0.08	1000–1056
Fe-P-B-C(1)	rest	1.0	7.2	1.3	1.62	0.16	954–1082
Fe-P-B-C(2)	rest	3.8	6.8	2.1	1.50	0.53	956–1189
Fe-P-B-C(3)	rest	1.3	7.1	2.6	1.21	0.23	956–1212
Fe-P-B-C(4)	rest	1.9	7.5	0.74	0.62	0.50	956–1040

^#^ Alloys may include Al, Cr, Ni, Cu, V and Ti in quantities of up to 0.2 wt% per individual element. * From Fe-P phase diagram in consideration of chemical compositions, as DSC was not able to analyse this alloy due to its high P content (quote by the device operator); melting range was checked by heating in a furnace in a Ar atmosphere by heating up to T = 1180 °C.

**Table 2 materials-16-03766-t002:** Hardness HV10 of alloys and microhardness HV of microstructural phases.

Alloy	HV10					Microhardness HV0.3 and 0.5 ^#^	
	Min.	Max.	SD *s*_H_	Mean HV/HRC	ΔHV	Eutectic	White Phases (Primary and Uneven Shapes)
Fe-C	450	492	15.1	470/45.5	42	639–714	348, 452 *
Fe-B	490	528	11.7	505/47	38	501, 543	1099, 1215
Fe-P(1)	565	606	13.8	589/52.5	41	677, 715	890, 927
Fe-P(2)	605	709	36.0	678/57	104	?	890, 999
Fe-P-C	620	774	53.3	715/59	154	835, 883	927, 966 ^(1)^
Fe-P-B(1)	678	789	40.2	718/59	111	746, 779	1174, 1391
Fe-P-B(2)	710	824	39.6	771/61.5	114	?	1043, 1119 ** 1324, 1391 **
Fe-P-B(3)	562	608	16.4	588/52.5	46	705, 746	441, 498 ***
Fe-P-B-C(1)	740	943	74.2	875/65	203	829, 855	966, 1025
Fe-P-B-C(2)	679	868	75.5	787/62	189	855, 883	1061, 1226
Fe-P-B-C(3)	742	857	41.5	805/62.5	115	842, 862	1129, 1261
Fe-P-B-C(4)	631	756	42.6	699/58	125	817, 883	974, 1025

^#^ Eutectic with HV0.3, intermetallic phases with HV0.5, lowest and highest values out of five measurements. * Pearlite. ** Lower values are a phosphorus phase, higher values are a boron phase. *** Solid solution, measured with HV0.05 due to small grains. ^(1)^ Uneven white shape field. ? Amount of eutectic was too small for measurements; however, we can conclude that it had the same hardness as the eutectic in the alloys from the same group.

## Data Availability

Data are available upon request.

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
