# Peer review of "Preliminary Study of New Low-Temperature Hard Abrasion Resistant Fe-P and Fe-P-X (X = C or/and B) Casting Alloys"

_materials, 2023, doi:10.3390/ma16103766_

Round 1
Reviewer 1 Report
The work analyzes the hardness and microstructure of Fe-P, Fe-B, Fe-P-C and Fe-P-B alloys. It is presented in a correct format and the structure is adequate, although it must improve certain aspects to consider its publication.
Although there is little bibliography referring to Fe-P-B alloys, a more in-depth review should be carried out, including works that carry out metallographic characterization.
For example, it should include:
Spyrydonova, I.M., Sukhova, O.V., Karpenko, N.V. et al. Application of Fe-P-B alloys in developing wear-resistant composite materials. J. Superhard Mater. 35, 97–104 (2013). https://doi.org/10.3103/S1063457613020056
The title indicates “new low-cost and low-temperature hard alloys Fe-P” however this alloy is not new. In addition, in the experimental part it is indicated that ferroalloys of FeP and FeB are used for manufacturing, so these alloys are already produced industrially.
The Production of alloys or Research methods should be improved incorporating the composition of precursor alloys, suppliers and specifying alloying temperatures, casting temperatures etc…. Only the melting temperature obtained by DSC analysis is indicated. The title of the paper indicates Synthesis, so the conditions of Synthesis have to be properly specified and discussed.
Although the DSC results are normalized by weight, the weight of samples used in the test should be indicated in the experimental conditions (the experimental conditions must be well defined).
Minor mistake Line 144, Ar 4.6 quality has a purity of 99.996 %
The peaks in the DSC scans should be further analyzed and discussed by comparing the data with those in the literature, and an attribution of the different peaks that appear should be made.
Only is indicated: “The lowest temperature 195 is always a low-temperature eutectic reaction while reactions occurring at higher temperatures are either higher-temperature eutectic reactions or reactions of various primary phases”.
In the microstructure analysis, the SEM images are at very low magnifications, so the qualitative carbon distribution is not adequately defined in many of the images. Considering the scale, only x500 magnification is used.
All EDS mapping should be performed at higher magnifications in order to clearly appreciate the distribution of C as well as the phases present.
Intermetallic compounds, although they can be seen clearly in the image, can be very thin due to the metallographic cut and, due to the volume of interaction of the electrons, a very different composition from the stoichiometric one can be obtained. A statistical analysis of the EDS analyzes should be included and detailed in the experimental design.
The complete EDS analysis must be provided for the identification of the phases. Phases are being identified indicating only the composition of P and B and as indicated, the EDS technique is semiquantitative. Therefore, the analysis of different eutectic zones/phases/eutectic phases, the average composition and the standard deviation of each of them should be included.
As it is indicated, the EDS technique is semiquantitative, therefore, the analysis of different phases, eutectics, etc. should be included. The average composition and the standard deviation of each of them should be indicated. The attribution of the phases must be carried out considering the average composition and the comparison with the bibliographic data.
Figure 4: It is indicated: Some intergranular eutectic is also visible in this alloy, Black lines, Figure 4b.
Perhaps, but the black lines do not justify their presence. It would be necessary to add an image at higher magnifications with the detail of the area.
Quizas, pero las lineas negras no justifican su presencia. Sería necesario incorporar una imagen a mayores aumentos con el detalle de la zona.
For the Fe-P-C sample, it is indicated that it consists of a ternary eutectic, again, it is necessary to provide an image at higher magnifications identifying all the phases present.
A basic description of the microstructure is made. The other metallographic analyzes present the same problem, the composition is given, but it is impossible to see the microstructure in the images provided at such low magnifications.
In general, the analysis of the microstructure is very poor and the discussion or analysis of results is reduced to listing phases without providing a rigorous analysis of it, and making the attribution simply considering the composition of P and B. There is no comparison of the results with the bibliography, especially, for the well-known phases.
The morphology of the phases and constituents should be discussed in more detail, clearly identified and labeled in the micrographs.
Regarding the hardness values, 5 hardness values are provided, the hardness values must be unique, and the mean value with the corresponding standard deviation must be indicated, making a statistical analysis to eliminate erroneous measurements.
The same for hardness of the phases, HV0.3 and HV0.5. Indicate mean value and standard deviation. The caption of the table 2 indicates HV0.01, It is this a mistake?
The difference in hardness is attributed to the different phases present and to “This is a logical consequence of a lower segregation rate and a more uniform distribution of the microstructural constituents”. (line 293)
What justifies a lower segregation? This should be discussed in more detail.
La diferencia de durezas se atribuye a la diferentes fases presentes y a “This is a logical consequence of a lower segregation rate and a more uniform distribution of the microstructural constituents”. ¿Que justifica una lower segregation? Esto debe comentarse con más detalle.
In the final part, it is indicated that the microstructure changes if homogenization and slow cooling are carried out, as would be expected, but an analysis of the compositional changes of the eutectic is not made, just a rough description, also indicating that it can be expected that change the mechanical properties of alloys.
This part should be eliminated or analyzed in more detail since as it is it does not provide information or novelty.
The last paragraph, before the conclusions, only discusses the use of etchants indicating that it would be necessary to use others.
I think this should be removed
It is not possible to present a “previous study of the microstructural analysis” with an unsuitable etchant when there are already previous metallographic analyzes that reveal the microstructure of alloys of similar composition.
What is the conclusion that is taken from the reading? We have carried out the metallographic study of the alloys with an inappropriate etchant and therefore the microstructure cannot be clearly appreciated and the results presented are not correctly analyzed. In the future we will try to do it properly.
For all these reasons, I consider that the work does not meet the adequate quality for its publication in its current form.
Author Response
Dear reviewer,
please see the attachement.
Kind regards,
Matija Zorc

Reviewer 2 Report
In this study, the authors performed preliminary investigations of the Fe-P and Fe-P-X (X = C or/and B) cast alloys. The influence of composition and cooling rate on the microstructure and hardness of the alloys were studied. The results are technologically sound and can provide some interesting insights into the microstructure and mechanical properties of Fe-P based cast alloys. However, some issues are still needed to be clarified before the manuscript can be accepted:
1. The title of the manuscript seems to be quite lengthy.
2. The Introduction section seems to be wordy. I suggest the authors try to condense it.
3. The Scientific background section seems unnecessary and should be folded into the Introduction section.
4. It seems that the Fe-P(1) and Fe-P-B(3) alloys are tougher than other alloys, as no cracks can be found around the corner of the indents for both alloys. The authors should give some explanations about this.
5. The authors claimed that the hardness of the alloy reaches values between 52 and 65 HRC depending on chemical composition and microstructure, showing their high wear resistance. However, there are other factors that affect the wear resistance of the alloys beyond merely the hardness. In fact, it has been increasingly recognized that coating elasticity and fracture toughness are also important factors affecting wear resistance, especially in abrasive, impact and erosive wear. Considering all the alloys are quite brittle, I hope the authors can put more words about the influence of brittleness on the wear resistance of the cast alloys.
The English expressions need to be further improved.
Author Response
Dear reviewer,
please see the attachment.
Kind regards,
Matija Zorc

Reviewer 3 Report
1. The abstract is necessary to improve, to include the method and results as solids and liquidus temperature and low-cost, i.e.
2. The citation number 40 on page 15 doesn´t appear in the references.
3. It is necessary to add in all figures DSC the sample quantity to do measurement and the place where it was obtained. Is it only one measurement or you did several? If only one measurement was done, then you must explain why this one represents the sample.
4. You must add ΔH in Figure's DSC to explain better your results.
5. With respect to the low cost in the title of this manuscript, it is necessary that you explain with an example how is the method to measure the cost. If this is not possible, please change the title.
6. What is the composition in purity for each metal? How we can be sure that there is no contamination into alloys?
7. I suggest including X-Ray Diffraction in Fig. 1b FeP-B-C (4) and Fig. 12a in order to demonstrate that several crystalline structures can exist.
Accept after minor changes
Author Response

(The authors gave the same response as above.)

Round 2
Reviewer 1 Report
The work has been improved with respect to the previous version but, from my point of view, not all the issues raised have been resolved.
The authors have removed “synthesis from the title”, possibly they misunderstood my previous comments. I just indicated that synthesis was included, and therefore the synthesis process should be indicated in more detail: composition of the precursor alloys and melting and casting temperatures, (section 3.1 has been improved in this version)
However, the title indicates “hard wear-resistance”. The work does not present wear resistance tests, the phases formed are very hard, but they are brittle (large cracks are seen in some of the microstructures) and, therefore, it would be necessary to carry out wear tests before indicating that they are alloys with high wear resistance
Minor mistake: ESXS and EDS is used to identify Energy-dispersive X-ray spectroscopy, please use only one.
DSC: The discussion of the results has not been improved, a very discreet analysis is made. Although it is indicated that there are no DSC data in the literature, the comparison with the bibliographic data can also be made since, as the authors indicate, the phases formed are well known and the melting or transformation points obtained should coincide.
These data should be referenced and compared with these data.
Microstructure: Authors indicate in author response that for some alloys “The carbon distribution in these alloys is global” and therefore it is not possible to obtain a distribution of the same through EDS, but In the manuscript is indicate the presence of Pearlite 0.76 wt% C and ledeburite consists of 4.3 wt% C.
The distribution of C is not appreciated due to the low magnification used in the EDS analysis.
If an adequate metallographic analysis is to be carried out, I cannot use the same magnification to see a structure with very large differences in size. Figure 3 shows the micro-constituents better in the optical microscopy image (at higher magnifications) than in the one obtained by SEM. This is a big mistake
The modification in figure 4 has a very low resolution, apart from this at low magnifications, perhaps x2000. With electron microscopy you can easily work at higher magnifications, perhaps at x10,000 you could see something.
Chemical composition: references to articles must be included indicating the similarity of the composition and morphology of the phases or microconstituents present.
Etchant: I did not indicate that the attack with Nital was inappropriate. The work indicates that (line 358) " An experiment with 358 Vilella’s reagent showed that other known etchants should be experimented with in the 359 future to reveal the microstructure more accurately"
It seems to me inappropriate to present a work on the microstructure of an alloy indicating that possibly with another attack the microstructure could be better seen and that it will be done in future research. Furthermore, considering that, as also indicated in the text, many of the phases/constituents are well known.
Hardness measurements HV10: I still insist on the need to include a statistical analysis of the hardness of the alloys.
Five hardness values are presented, Why? How many hardness measurements have been made to obtain each of the five values?
Each hardness value should correspond to the mean of several hardness measurements made in similar areas of the sample and therefore all of them should have a standard deviation.
What standard was followed to carry out the measurements? All hardness standards indicate that the hardness results must be presented with the corresponding uncertainty in the hardness determination
A correct analysis can be presented with the mean and the standard deviation applying the corresponding statistical analysis that eliminates the erroneous measurements.
It can also be done with min max (mean, median, and mode) or present the data with a box and whisker plot and the corresponding quartiles.
I can assume that 20 hardness measurements were made, and five values are presented with the minimum maximum median values and those corresponding to 25% and 75%. Presenting five hardness values without indicating that they represent does not have any scientific rigor.
For all these reasons, I consider that the work still does not meet the adequate quality for its publication in Materials Journal in its current form.
Author Response
Dear reviewer,
please see attachment for our comments.
Yours sincerely,
Matija Zorc
